# Exploiting Vehicular Social Networks and Dynamic Clustering to Enhance Urban Mobility Management

**DOI:** 10.3390/s19163558

**Published:** 2019-08-15

**Authors:** Ademar Takeo Akabane, Roger Immich, Richard Wenner Pazzi, Edmundo Roberto Mauro Madeira, Leandro Aparecido Villas

**Affiliations:** 1Institute of Computing (IC), University of Campinas (UNICAMP), 1251 Albert Einstein Av., Campinas, SP 13083, Brazil; 2Faculty of Business and Information Technology (FBIT), Ontario Tech University, 2000 Simcoe St N, Oshawa, ON L1H 7K4, Canada

**Keywords:** vehicular social networks, dynamic clustering, urban mobility management, social network analysis, social network concepts, advanced traffic management system

## Abstract

Transport authorities are employing advanced traffic management system (ATMS) to improve vehicular traffic management efficiency. ATMS currently uses intelligent traffic lights and sensors distributed along the roads to achieve its goals. Furthermore, there are other promising technologies that can be applied more efficiently in place of the abovementioned ones, such as vehicular networks and 5G. In ATMS, the centralized approach to detect congestion and calculate alternative routes is one of the most adopted because of the difficulty of selecting the most appropriate vehicles in highly dynamic networks. The advantage of this approach is that it takes into consideration the scenario to its full extent at every execution. On the other hand, the distributed solution needs to previously segment the entire scenario to select the vehicles. Additionally, such solutions suggest alternative routes in a selfish fashion, which can lead to secondary congestions. These open issues have inspired the proposal of a distributed system of urban mobility management based on a collaborative approach in vehicular social networks (VSNs), named SOPHIA. The VSN paradigm has emerged from the integration of mobile communication devices and their social relationships in the vehicular environment. Therefore, social network analysis (SNA) and social network concepts (SNC) are two approaches that can be explored in VSNs. Our proposed solution adopts both SNA and SNC approaches for alternative route-planning in a collaborative way. Additionally, we used dynamic clustering to select the most appropriate vehicles in a distributed manner. Simulation results confirmed that the combined use of SNA, SNC, and dynamic clustering, in the vehicular environment, have great potential in increasing system scalability as well as improving urban mobility management efficiency.

## 1. Introduction

Over recent years, the research community in the field of communication and ad-hoc networks has been very attracted to social network analysis (SNA) and social network concepts (SNC) to design and implement new algorithms and protocols for socially aware networks, such as mobile social networks (MSNs) and vehicular social networks (VSNs). The legacy of social networks in communication networks is that all entities have a certain degree of interdependence to each other [1]. Such interdependencies can include network topology similarity, physical contact, community, and mutual interest. In addition to the interdependencies, the correlations between the entities can be explored in SNA. Social networks are a virtual group of entities that have some social interdependencies among them, and such interdependencies can be applied to improve the efficiency and effectiveness of network services [2,3].

The VSN paradigm has emerged through the integration of the concepts of mobile social networks and vehicular ad-hoc networks (VANETs) [4,5,6]. As a consequence of this integration, two approaches can be explored in the vehicular environment, such as (i) application of the SNA [5,6] techniques and/or (ii) use of the SNC [4,6]. The first approach focuses on identifying the node importance in the network. To this end, three main measures of centrality most used in VSNs are degree, closeness, and betweenness [5,6]. It is known that the network topology, in VSNs, is highly dynamic and consequently calculating the node centrality is a challenging task. On the other hand, once identified, it can be useful for many applications such as the management of information flow in the network. The second one, however, involves social interactions between nodes that have mutual interests in the temporal virtual community [4,7]. In other words, such an approach provides the opportunity of vehicles to participate in a virtual vehicle community and share information of mutual interest through social interactions. Based on this idea, each vehicle can share their social information, for example, the personal route. In this way, allowing the practice of collaborative route-planning. The social interaction occurs when vehicles meet each other and share their social information through wireless communication.

The highly dynamic topology of VSNs is due to the high mobility of the nodes (or vehicles), thereby the development of systems in such networks must take into account this characteristic. For example, due to high mobility, the communication links between vehicles only last for short periods of time [8]. Two communication types most commonly applied in VSNs are vehicle-to-vehicle (V2V) and vehicle-to-infrastructure (V2I) [9]. The IEEE 802.11p WLAN standards have been used as the dedicated frequency spectrum for inter-vehicular communication in the 5.9 GHz band and have also assigned multiple communication channels, such as control channel (CCH) and service channels (SCHs).

ATMS integrates communication, storage, and processing technologies to collect raw data from the VSNs, to extract knowledge of vehicular traffic on roads [7]. The ATMS can provide services to improve traffic management efficiency and safety using such knowledge. For better performance, many ATMS applications require vehicles to periodically share their data (floating car data) between neighboring vehicles, a central server, and/or roadside unit (RSU). Through this sharing, it is possible to create awareness about vehicular traffic conditions [10,11,12]. This practice is known as beaconing and the data exchanged is associated with vehicle mobility, such as vehicle identification, current vehicle position, speed, the direction of travel, just to name a few examples. This data exchange is performed by the CCH and generally at a transmission frequency between 1 Hz and 10 Hz [13].

Different ATMSs have been designed and implemented to overcome the lack of urban mobility that affects the daily life and well-being of the citizens [10,11,12,14]. Several solutions implement a centralized approach [11,12] due to the difficulty of selecting the most appropriate vehicles, in highly dynamic networks, for congestion detection and calculation of alternative routes. As a result, such solutions are not easily scalable. Another solution employs a distributed approach for congestion detection and calculation of alternative routes [14]. However, to achieve its goal, such a solution needs to segment the entire scenario into multiple sub-regions beforehand. Moreover, the alternative route is calculated selfishly, i.e., without considering the routes chosen by neighboring vehicles.

Based on the gaps found, SOPHIA, a distributed **S**ystem of urban m**O**bility management based on a collaborative a**P**proach in ve**HI**cular soci**A**l networks was designed and implemented. Inspired by the two VSN approaches mentioned above, an SNA technique to classify and select the vehicles in each clustering was applied to reduce bandwidth consumption. Two SNCs were employed (social interaction and virtual temporal community) to perform the exchange of information of common interest. This exchange of information helps in alternative route-planning in a collaborative way, thus improving urban mobility management.

In brief, the focus of the SOPHIA system is to minimize the problems associated with traffic congestion, in a distributed manner, and without jeopardizing its scalability. The evaluation of the proposed system was performed through simulations comparing with other systems of the literature [11,12,14] in different requirements. Additionally, several performance assessments were conducted following three perspectives: (i) control channel assessment, (ii) scalability assessment, and (iii) traffic management assessment.

The main contributions of this paper can be summarized as follows:A novel dynamic clustering approach based on SNA along with received signal strength (Section 3.2). This approach is applied to improve the data flow within the network;A novel collaborative rerouting approach based on social interaction and virtual temporal community to enhance urban mobility management (Section 3.4).

The remainder of this paper is organized as follows. In Section 2, an overview of existing work in the literature about urban mobility management is described. The proposed solution is presented in Section 3. The simulation setup and experimental results are given in Section 4. Finally, conclusion and future work are presented in Section 5.

## 2. Related Work

This section presents the related works relevant to the design and implementation of SOPHIA system. Moreover, the aspects related to dynamic clustering algorithms are discussed along with infrastructure-less and infrastructure-based for urban mobility management.

### 2.1. Dynamic Clustering Algorithms

Grouping nodes into clusters has been extensively investigated in many fields, such as wireless ad-hoc networks and mobile ad-hoc networks, by focusing mainly on energy saving [15,16,17]. In VANETs, due to the high topology changes, the clustering algorithms proposed for other kinds of ad-hoc networks such as mobile sensor networks are not suitable to be applied in VANETs [16].

In VANETs, clustering techniques have been proposed to improve communication efficiency and facilitate network management, by grouping vehicles in a geographical vicinity together. The advantages of clustering can be visible in highly dynamic networks, in which information aggregation and management can be performed in each network cluster [18]. Thus, clustering can increase the network scalability and decrease the communication overhead.

Hafeez et al. [19] proposed a clustering algorithm by considering speed as the main parameter to build clusters. The cluster head (CH) is elected in a distributed manner according to their relative speed and distance from their cluster members (CMs). This algorithm improves cluster stability through diffuse speed processing. Besides that, it chooses the second optimal vehicle as the temporary CH when the original one becomes unavailable.

In [20], the authors proposed a mobility-based clustering scheme according to the parameters of the vehicle’s movements, such as moving direction, relative velocity, and the relative distance between vehicles. Such parameters are applied to select the CH. In mobility-based clustering, each CH is located in the geographical center of a cluster, and CMs are inside transmission range of the CH and moving in the same direction as the CH. Hassanabad et al. [21] also proposed a mobility-based clustering scheme like the aforementioned one. The difference between them is that the latter applies the Affinity Propagation algorithm, proposed by the authors, to produce clusters with high stability.

Abuashour and Kadoch [22] proposed the algorithm named CORA—Control Overhead Reduction Algorithm. The proposed algorithm aims to minimize the overhead network generated by CMs in a clustered segment scenario. The CHs are selected based on maximum lifetime among all vehicles that are located within each cluster.

### 2.2. Infrastructure-Based Urban Mobility Management

In [11], the authors proposed a centralized system for traffic management called EcoTrec. The proposed system is centralized because of congestion detection and alternative route calculation are performed by a central entity. The EcoTrec system aims to reduce CO_2_ emissions without significantly increase travel time. To this end, the system was built on a three-component architecture: Vehicle Model, Road Model, and Traffic Model. The Vehicle Model collects and updates the individual information of the vehicle, as well as periodically sharing them with the Road Model. The shared information comes from Global Positioning System (GPS), accelerometer, and gyroscope embedded in vehicles. The Road Model is hosted in the RSUs which are along the roads and connected by the Traffic Model. The Traffic Model is a central server containing the characteristics and road traffic conditions. Both Road Model and Traffic Model communicate with vehicles through V2I communication. Each vehicle makes periodic requests to the server about the road traffic condition and if the route is congested, the server sends an alternate route.

In [10], the authors introduced Next Road Rerouting (NRR). The main objective is to assist drivers in choosing the next most appropriate road, to circumvent the congested areas. The proposed system operates in two-stage traffic management: (i) estimates only the next road for the vehicle to bypass the congested point, and thereafter, (ii) uses the vehicle’s GPS to calculate the remainder of the alternate route to the destination. The reason for this approach lies in the fact that the calculation of the next road is less costly than the recalculation of the end-to-end route. The NRR mechanism needs a central server (Traffic Operation Center) to gather all the traffic information. In this case, NRR assumes that there is a traffic light at each intersection, to collect such information. Once the congestion is detected, the server notifies the nearest traffic light of the congested area. Thereafter, the traffic light notifies the next most appropriate road for vehicles. After that, the rest of the route is calculated with the aid of the vehicles’ GPS.

Pan et al. [12], the authors proposed a hybrid urban vehicle management system named DIVERT. It is considered a hybrid approach because it requires a central server to collect information from vehicles and detect vehicular traffic condition. The alternative routes calculation is carried out by the vehicles in a collaborative manner. In the DIVERT system, the central server operates as a coordinator that receives the vehicle information (speed, location, and direction) via V2I communication. Through this information, the server can detect congested locations and inform the vehicles that are driving to such locations. In this system, the responsibility for the alternative routes calculation is given to the vehicles. Once they need to compute an alternative route, it must take into account the chosen route of the neighboring vehicles, i.e., a collaborative routing decision applies. It is important to notice that in the DIVERT system, the broadcast suppression mechanism was not applied during the message dissemination process. This can lead to a broadcast storm problem.

### 2.3. Infrastructure-Less Urban Mobility Management

In [14], the authors proposed a distributed system for vehicular traffic management, named FASTER. In the proposed system the congestion detection and alternative route calculation do not need any infrastructure. To achieve its goal, FASTER needs to previously segment the entire scenario into multiple sub-regions (or districts). This is performed to aggregate traffic information. Each district has an area equal to 1-hop communication. Each vehicle periodically collects and transmits information, such as average speed and route identification to everyone within its transmission range. The vehicle closest to the center of the district is selected to initiate the dissemination of traffic information aggregated to other vehicles. During the dissemination process, a broadcast suppression mechanism is applied to avoid network overhead. In such a system, the calculation of the alternative route is performed selfishly, based on the probabilistic *k*-shortest path.

Kasprzok et al. [23] presented a decentralized congestion avoidance strategy for connected vehicles. Their approach measures the vehicular traffic congestion level of a road segment using the amount of wireless network traffic generated by vehicle-to-vehicle communications. The vehicle computes an alternative path employing a modified *k*-shortest path algorithm whose paths are weighted using a Logit model [24] upon the congestion is detected.

In [25], the authors proposed a fully distributed congestion avoidance system which detects traffic congestion and reroutes vehicles to minimize their travel time. The proposed system does not require global traffic information to detect congested areas but rather only the local information about the traffic conditions. According to local traffic information, each vehicle computes the traffic condition in its current road segment. Hereafter, if necessary, it requests information about the alternative paths of the surrounding vehicles to make the choice that will minimize its remaining travel time. This system relies on sending information request messages whenever a vehicle desires or needs to know more about upcoming roads and traffic. This strategy was applied to reduce network overhead and increase system scalability.

On one hand, infrastructure-based vehicular traffic management systems have been most explored, due to the difficult task of selecting the most relevant vehicles within a subset for detecting congestion and calculating alternative routes. On the other hand, distributed systems cannot ignore such a task. For this, for example, in work of [14] is previously segmented the entire scenario and the most central vehicle is chosen. However, this choice is not always the most appropriate. To overcome this gap, a novel dynamic clustering approach based on SNA along with received signal strength was proposed. In addition, most of the known solutions suggest alternative routes in a selfish fashion. To overcome this gap, a novel collaborative rerouting approach based on social interaction and virtual temporal community was proposed.

## 3. Towards the Design of SOPHIA

SOPHIA is a distributed system for urban mobility management based on a collaborative approach in vehicular social networks. The aim of such a system is to improve the vehicular flow on the roads without compromising the system’s scalability. Taking this into consideration, the system is composed of four components: (i) vehicular crowdsensing; (ii) dynamic clustering approach; (iii) knowledge extraction and distribution; and (iv) collaborative route-planning. Details of each component are presented below.

### 3.1. Vehicular Crowdsensing

The mobile crowdsensing paradigm (MCS) employs the concept of ubiquitous computing in the collection and sharing of data [26,27]. In other words, this paradigm aims to incentivize participants to efficiently and effectively contribute to a common goal to use context-related sensing data from their mobile devices in solving a specific problem in a collaborative manner [28]. In addition, by aggregating the crowd-generated local data, it is possible to create cooperative local awareness. Such awareness can lead to improvements in numerous large-scale applications, such as air pollution monitoring and traffic congestion warnings. Since vehicles are equipped with wireless communication technologies along with smart sensors in VSNs, that enables the vehicle crowdsensing (VCS) paradigm [27]. This paradigm, in turn, enables the monitoring of dynamic and large-scale phenomena [29].

The motivation for using VCS lies in the fact that the participants of the networks can solve problems in cooperation. For example, VSN participants can jointly improve urban mobility by sharing data collected about traffic conditions. In doing so, VSNs’ systems can aggregate the collected data and extract knowledge (local awareness) about real-time traffic conditions. Thus, the knowledge extracted can assist in urban mobility management.

In this work, the VCS paradigm was applied to create the local traffic awareness, in which vehicles cooperate to sense and collect urban data requested by the system. For this purpose, it was assumed that each vehicle (*n*) periodically generates a packet ($b_{n}$) containing some data collected from onboard units, such as current speed ($s_{n}$), location ($p_{n}$), time stamp ($t_{n}$), and vehicle score ($v_{esc_{n}}$), as described in Equation (1). The $v_{esc_{n}}$ will be used in the dynamic clustering mechanism which will be explained later.

(1)$$b_{n}=\left(p_{n},s_{n},v_{esc_{n}},t_{n}\right)$$

### 3.2. Dynamic Clustering Approach

One of the great challenges in highly dynamic networks is to select the most appropriate nodes within a subset to perform a given task [16]. A straight solution for this problem is to employ an infrastructural approach, for example, RSUs and/or a central server, [10,11,12], thus eliminating the difficult task of selecting vehicles. To overcome this challenge in an infrastructure-less approach, the proposed work adopts a dynamic clustering technique. Unlike the FASTER [14] system, SOPHIA does not need to segment scenario to select the most appropriate vehicle that will perform the congestion detection task.

Network clustering is the division of a graph into a set of subgraphs, called clusters. Each cluster elects one node leader (CH), according to some rules, that works as a local management entity. In addition to that, CMs are all nodes from CH’s 1-hop neighbor set. A 1-hop cluster is a clustering such that every node in the network can communicate in 1-hop with the CH of the cluster it belongs to. The cluster is composed of two levels of communications [20]. The first one is intra-cluster communication, where CMs can directly communicate with its CH or nearby CMs within the same cluster. The second one is when a CH communicates with nearby CHs or roadside infrastructures, which is known as inter-cluster communication.

As a general procedure in cluster formation, the nodes participating in, or seeking to join in one, will typically carry out some or all the steps described below [16]:1.*Neighborhood discovery*: a node generally announces its existence to its neighbors through a periodic short-message transmission, while simultaneously gathering the same message from its neighbors;2.*CH selection*: after collecting data about the local environment, each node will compute, based on some rule, to find the most appropriate node to act as its CH. In this step, the node can also consider its suitability to be a CH;3.*Affiliation*: the node will contact the neighbor node that was chosen as the appropriate CH and seek to become a CM of that cluster;4.*Announcement*: the most appropriate CH may then send an announcement message to its neighbors to initiate the process of cluster formation;5.*Maintenance*: this step is divided into two parts:(a)*As a CH*: if a CH loses all connections with its CMs, the cluster is assumed to be dead, and the procedure is started again (Step 1). On the other hand, a cluster can merge with another one and become a larger cluster. In this case, the node will execute the Step 5(b); (b)*As a CM*: the node periodically evaluates the link to its CH. If the link fails it will return to Step 1. If the node receives an affiliation request from a node that does not belong to its group, it can start the CH selection again (Step 2) to choose the next appropriate CH.

In SOPHIA, each cluster is associated with a set of vehicles called CMs and a representative of CH, as shown in Figure 1. The vehicles depicted by the labels *A* and *B* represent the CHs of the clusters 1 and 2, respectively, while the other vehicles portray the CMs. The vehicle label as 1 will be used in an example afterward. The CH is the vehicle temporarily selected with the responsibility of gathering and forwarding the information on behalf of the CMs. The vehicle with the highest score ($v_{esc_{n}}$) is selected as CH, the details of the scoring computation are given below. By means of the dynamic clustering approach, it is possible to overcome the following challenges: (i) selecting the most appropriate vehicle in a distributed manner; (ii) minimizing the network overhead; (iii) increasing the scalability of the system; and (iv) facilitating the data flow within network. It is noteworthy that in congested areas, fatally, there will be vehicles in multiple clusters and this particularity was explored to improve the flow of data on the network, otherwise, the information flow would be interrupted.

Our dynamic clustering algorithm procedure only takes into consideration Steps 1 and 2 of the aforementioned general procedure. The idea here is to explore the social properties of nodes to select the CH to improve data flow in the network. This improvement can be done by a path with minimal interference in communication along with the social properties of nodes. To achieve this goal, each vehicle autonomously calculates its score according to neighborhood communication links. This calculation performed together with a received signal strength indicator, as shown in Equation (Equation 2).

(2)$$v_{esc_{n}}=\sum_{\begin{array}{c} M_{n}\left(i,j\right)\neq 0,i<j \end{array}}\frac{1}{M_{n}^{2}{\left[1-M_{n}\right]}_{i,j}}+\left(PL\left(d_{0}\right)+10\alpha \log\left(\frac{d}{d_{0}}\right)+\Psi_{\sigma_{single}}\right),d\geq d_{0}$$

For simplicity’s sake, initially, let us focus on only the first half of the equation. This part of the equation describes an egocentric network metric. One advantage of this metric is that it applies only the locally available topology information. More specifically, the egocentric betweenness metric (EBM) [30,31]. EBM aims to indicate the relevance of the node for the information flow continuity in the network. It is known that an adjacency matrix, ($M_{k\times k}$), can represent the intercommunication links between the nodes, in which *k* is the number of 1-hop communication. The EBM calculation is given by the inverse sum of the equation ($M_{n}^{2}{\left[1-M_{n}\right]}_{i,j}$), where $M_{n}$ denotes the adjacency matrix of the vehicle *n*, $M_{n}^{2}$ represents the geodesic distance between the pairs of vehicles *i* and *j*, and finally, 1 in the expression corresponds to a matrix with all elements equal to 1.

The second half of the equation refers to the received signal strength indicator. The log-distance path-loss [32] was the model applied. *d* is the Euclidean distance between vehicles, $d_{0}$ is the distance from a reference point to the emitter, $PL(d_{0})$ is the power of the reference point to the sender, α describes the path-loss exponent (it varies according to the environment), and $\Psi_{\sigma_{single}}$ is a variable that describes the attenuation of the communication signal. In brief, the power of the received signal fades logarithmically with the distance between the vehicles.

For each change in the local topology, the vehicle’s score should be updated. Algorithm 1 describes the procedure of our proposed dynamic clustering. For every change in the network topology (Lines 2 to 4), which corresponds to Step 1 of the general procedure in cluster formation, the vehicle score is recalculated (Line 7), which matches the Step 2. Thereafter, the value is added to $v_{esc_{n}}$ and transmitted in the subsequent beacon package ($b_{n}$), Line 10.

**Algorithm 1:** Vehicle score calculation.  **inputs:**
*N* = {*n*_1_, *n*_2_, ..., *n_n_*} the set of all vehicles that are currently within the transmission range  **output:** Vehicle score (*v_escn_*)

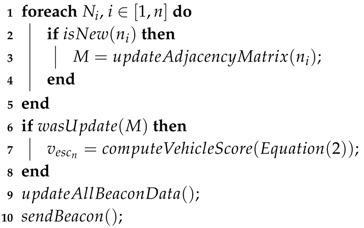



### 3.3. Knowledge Extraction and Distribution

To better understand the details of the aggregation functions for knowledge extraction, a formal definition of the road network topology is required.

**Definition** **1.**
*The road topology can be represented through a directed graph G=(V,E,W), where V corresponds to a set of intersections (v), whereas E denotes to a set of segments (e, where $e\in E\subseteq V^{2}$). In addition to that, a set of weight (ρ∈W) is attributed to each road segment. This weight indicates the level of service and will be explained in detail later on. Finally, a route between two points A and B, r(A,B), is a sequence of intersections $\left(v_{1},\ldots ,v_{n}\right)$ such that $v_{1}=A$, $v_{n}=B$ and all pairs of consecutive intersections are connected by a road segment, i.e., for all i=1,…,n-1 exists $(v_{i},v_{i+1})\in E$.*


To extract the knowledge about the vehicular traffic condition, two different aggregation functions are required, i.e., (i) aggregation of beacons received from the neighborhood—local awareness (Equation (Equation 3)) and (ii) aggregation of local awareness—knowledge of the traffic condition (Equation (Equation 4)).
(3)$$\Lambda :=(E^{\prime },Y,\Omega )$$
where $E^{\prime }=\left\{e_{1},\ldots ,e_{n}\right\}|E^{\prime }\in E\left(G\right)$. The parameters Y and Ω are $\left\{t_{1},\ldots ,t_{n}\right\}$ and $\left\{v_{m_{1}},\ldots ,v_{m_{n}}\right\}$, i.e., the current time and average speed of each element of $E^{\prime }$.
(4)$$\Lambda_{r,s}:=\sum_{\begin{array}{c} 1\leq r,s\leq n \end{array}}\sigma \Lambda_{r}+\left(1-\sigma \right)\Lambda_{s}\;\;,\;\;\left\{\begin{array}{c} t_{r}>t_{s}\\ s_{r}\;,\;s_{s}\neq 0 \end{array}\right.$$
where σ is the weighting factor. The purpose of such a factor is to consider the most current information in the information aggregation process ($t_{r}>t_{s}$).

Considering again the example of the Figure 1, assuming that the vehicle 1 starts the process of extracting local awareness. After finishing the initial process, it forwards the local awareness to the CH (vehicle *A*) of its cluster. After that, the CH performs the aggregation of the beacons of its neighborhood (Equation (Equation 3)) and the aggregate information received from the vehicle 1 (Equation (Equation 4)). The result of that will be forward to the subsequent CH (vehicle *B*) until it reaches the vehicle with the highest score. In this example, the vehicle *B* has the highest score temporarily, therefore, such a vehicle is responsible for computing the weight of each road segment according to Equation (Equation 5).
(5)$$\rho_{k}=v_{agr_{k}}^{avg}\times {\left(1-v_{e_{k}}^{lim}\right)}^{-1}|\forall e_{k}\in E^{\prime }$$
where the parameters $v_{agr_{k}}^{avg}$ and $v_{e_{k}}^{lim}$ correspond the average aggregate speed and the maximum speed allowed on the road, *k*, respectively.

After this step, the vehicle *B* classifies the weight of the road segment according to the level of service (LOS) according to Table 1. This table shows the traffic classification for each service level according to the weight (ρ) calculated by Equation (Equation 5). Each service level depicts a traffic condition. If during the classification process, the LOS *D*, *E*, and *F* are found, a message containing identification about these roads segment is generated and the dissemination process begins. To avoid the problem of the broadcast storm during the data dissemination process, the concept of preference zone (ZoP) [33] was applied. ZoP is a region within the transmission range, whose vehicles within it are most proper to continue the dissemination process. The ZoP concept is based on the delay, this means that the vehicles within it have lower delay (or priority) than the vehicles outside it. Thus, vehicles outside the ZoP receive redundant messages and cancel the scheduled transmission.

### 3.4. Collaborative Route-Planning

As mentioned earlier, VSNs involve social interactions (also known as social object relationship—SOR [35]) within a temporal virtual community of vehicles based on common interests or mutual goals [4,6]. The common interests applied in this work is the alternative routes chosen neighborhood vehicles. Inspired by this idea, it was proposed the collaborative route-planning employing two SNC concepts, such as temporal virtual community and social interactions, as shown in Figure 2. Therefore, all vehicles within the temporal virtual community area are considered participants of such a community. The social interactions between community participants are realized through V2V communication and the information of common interest exchanged are the alternative routes chosen. It is worth mentioning that the area covered by the temporal virtual community depends on the circumference radius defined by the application, and the location of the congestion point was defined the central point of the community area. The main goal in this step is to route vehicles away from the current congestion point, without creating secondary congestion points.

For the sake of clarity, Algorithm 2 is introduced, which describes the procedure of collaborative route-planning. During the route-planning phase, vehicles within the temporal virtual community and closest to the congestion point have priority in choosing an alternative route, i.e., they have the shortest waiting time in choosing an alternative route. This time is directly proportional to the distance between vehicle and congestion point (Line 1). Before calculating an alternative route, the vehicle computes the road popularity (pop) according to the alternative routes chosen by the neighborhood vehicles (Line 3).The pop indicates the most popular roads chosen by vehicles to bypass congestion areas. Thus, road popularity (*v*) is given by Equation (Equation 6).
(6)$$pop_{v}=num_{v}\times \left(\frac{len_{\left(v\right)}}{lin_{\left(v\right)}}\right)$$
where $num_{v}$, $len_{\left(v\right)}$ and $lin_{\left(v\right)}$ represent number of vehicles, road length, and lines on the road surface, respectively.

**Algorithm 2:** Collaborative route-planning for vehicles that are moving toward the congested road.  **inputs:**
*msg*—warning message, which contains the coordinates of the traffic congestion point (*s_x_*, *s_y_*). (*r_x_*, *r_y_*) depicts the coordinates of the receiving vehicle  **output:**
*r* - the alternative route chosen

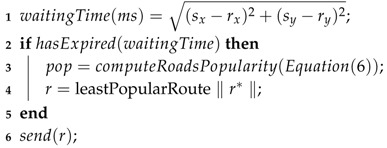



Now suppose that $r^{*}\left(p_{cur},d_{est}\right)$ denotes the set of all possible alternative routes from the current position ($p_{cur}$) to the destination ($d_{est}$). Thus, the choice of an alternative route is given by Equation (Equation 7), in other words, the vehicle selects the least popular route (*r*) among all possible routes (Line 4) and shares it through social interaction (Line 6). In this way, reducing the possibility of generating congestion points in another place in the near future.

(7)$$r=\underset{r\in R(p_{cur},d_{est})}{\mathrm{leastPopularRoute}}\left\|r^{*}\right\|$$

## 4. Performance Evaluation and Results

This section shows the performance assessment of SOPHIA and compares it to FASTER [14], DIVERT [12], and EcoTrec [11] systems. In addition, the EcoTrec system is going to be used as a baseline due to its simplicity. It is worth mentioning that SOPHIA’s aim is to make the most of public roads without compromising the system’s scalability. For a better presentation, this section was divided into four subsections: simulation setup is shown in Section 4.1 and the results and analysis of simulations were divided into: control channel assessment—Section 4.2, scalability assessment—Section 4.3, and traffic management assessment—Section 4.4.

### 4.1. Simulation Setup

The performance evaluation of SOPHIA was conducted through simulations using the Veins 4.5—Vehicular Network Simulations (http://veins.car2x.org/). This is an open source framework which integrates two simulators well-known tools, i.e., OMNet++ 5.0—Network Simulation Framework (https://omnetpp.org/) and SUMO 0.29.0—Simulation of Urban MObility (http://sumo.sourceforge.net/). In addition, the Handbook Emission Factors for Road Transport (HBEFA, http://www.sumo.dlr.de/wiki/Models/Emissions/HBEFA-based) was used to measure CO_2_ emissions. It is worth remembering that HBEFA is natively implemented in SUMO.

The TAPASCologne project (http://kolntrace.project.citi-lab.fr/) of the Institute of Transportation Systems at the German Aerospace Center (ITS-DLR) was adopted in the simulation process. This project aims to reproduce the vehicle traffic, with the highest possible level of realism, in a large-scale scenario of the city of Köln, Germany, see Figure 3.

We chose the dataset that contains traffic data traces from 6:00 a.m. to 8:00 a.m., representing more than 250,000 vehicle routes. However, only a central submap was chosen for the simulation experiments because it displays a higher incidence of traffic congestion (LOS D, E, and F—heat bar), as shown in Figure 3. With the traffic demand of the submap, it was constructed a new dataset (containing more than 46,000 vehicles routes) and divided into five different vehicle insertion rates, namely 20%, 40%, 60%, 80%, and 100%. For example, 20% means that only 20% of the total vehicles are inserted in the scenario for the simulation experiments, and so on. All the experimental results of this work were conducted with a confidence interval of 95%. Table 2 summarizes the simulation parameter settings.

Additionally, nine metrics were used to evaluate the performance of the SOPHIA system. These metrics were divided into three perspectives (or assessments), which are described in detail below.

Control channel assessment**Channel busy ratio**: indicates the interference level. This is estimated as the fraction of the time in which the channel is identified as busy due to packet collisions or successful transmission;Scalability assessment**Overhead**: measures the total amount of transmitted messages by the vehicles;**Latency**: demonstrates the time spent to deliver the messages to the vehicles;**Packet loss**: shows the total number of lost packets during the message transmissions;**Coverage**: indicates the percentage of messages successfully delivered.Traffic management assessment**Travel time**: indicates the average travel time in relation to all vehicles;**Travel Time Index**: measures the level of urban traffic congestion [36]. This index is calculated by the ratio of the total travel time to the free-flow travel time;**Congestion time loss**: describes the average time spent on congestion;**CO_2_ emission**: gives the average CO_2_ emission of all vehicles.

### 4.2. Control Channel Assessment

As all the solutions apply the beaconing approach in their solution to achieve the goals, and the channel used for that purpose is the control channel. Then, the assessment of the control channel is necessary to analyze. In the experiments, the beacon transmission rate of 1Hz was set to all systems.

Figure 4 shows the performance result of the control channel in relation to the vehicle insertion rate. The table (top of figure) depicts the channel busy ratio while the bar chart (bottom) depicts the gain over EcoTrec. As expected, the channel busy ratio increases with the vehicle insertion rate because of the number of vehicles in the neighborhood increases, thereby raising the competition for control channel access. Among all the analyzed solutions, SOPHIA has the lowest average channel busy ratio for all vehicle insertion rates. The reason for this behavior is due to the system’s ability to perform better vehicular traffic management. In a few words, SOPHIA distributes vehicular traffic to make the most of the availability of public roads. As a result, the homogeneous distribution of vehicular traffic on the roads reduces the consume on the control channel bandwidth. In addition, we can observe that SOPHIA, FASTER, and DIVERT have a gain, on average, of 19%, 15%, and 11.17%, respectively, over EcoTrec in all vehicle insertion rate. It is important to notice that, on average, SOPHIA had 27% better result in comparison to FASTER and a 70% improvement in comparison to DIVERT.

### 4.3. Scalability Assessment

This subsection analyzes the scalability results of SOPHIA against the FASTER, DIVERT, and EcoTrec systems in terms of overhead, packet loss, latency, and coverage metrics. The results are displayed in Figure 5. Each sub-figure is composed of two bar charts. The top one represents the numerical value of the assessed metric and the bottom one represents the gain with respect to EcoTrec. Figure 5a displays the performance results of all the evaluated systems according to the overhead metric. Both systems, EcoTrec and DIVERT, constantly need to exchange messages between the vehicles and the central server to reach their purposes. Due to this strategy, it is possible to observe that both have a higher average rate of messages transmitted in relation to FASTER and SOPHIA. Another determining factor for this high rate, for both systems, is the absence of a broadcast suppression mechanism during the message distribution process. By examining carefully, it is possible to notice that DIVERT has a slightly higher transmission rate than EcoTrec. This is because DIVERT, in addition to communicating with the central server, implements a collaborative routing mechanism when choosing an alternative route. It is worth mentioning that such a mechanism contributes to vehicular traffic management and this contribution will be discussed in the following subsection. Both FASTER and SOPHIA apply vehicle selection techniques for the extraction of knowledge. FASTER segments the scenario into several sub-regions and in each of them one vehicle for knowledge extraction is selected. However, SOPHIA applies a dynamic clustering approach to select the most appropriate vehicle. The dynamic clustering is more appropriate, in this case, as it does not need to segment the scenario for the vehicle selection. It should also be mentioned that both FASTER and SOPHIA applies a mechanism to deal with the broadcast storm problem. Additionally, both have similar performance and they can drastically reduce the total amount of transmitted messages, more than 91% decrease in comparison with DIVERT and EcoTrec, as shown in Figure 5a (bottom).

Figure 5b shows the number of packet loss according to the vehicle insertion rate. Since it is known that EcoTrec and DIVERT systems have the highest network overhead compared to FASTER and SOPHIA (Figure 5a), it is expected that both also have similar results in relation to the packet loss metric. This expectation is confirmed in Figure 5b. It shows that solutions that have higher transmission rates also have a greater amount of packet loss. Since FASTER and SOPHIA have the lowest network overhead among its competitors, consequently they also have lower packet loss rates. Another factor that causes the rising of packet loss is the intermittent connection between vehicles. According to Figure 5b (bottom), the percentage reduction achieved for FASTER and SOPHIA is around 70.8% and 74.2% for all vehicle insertion rates, compared to EcoTrec and DIVERT, respectively.

Another metric evaluated is the transmission latency in relation to the vehicle insertion rate, Figure 5c. In both, the infrastructural and distributed approaches, as vehicle insertion rates increase the latency also increases, as expected. This is because raising the number of vehicles in the simulations increases the network overhead caused by the exchange of messages. However, FASTER and SOPHIA have the lowest latencies compared to other systems analyzed. Comparing numerically, the mean delay of the SOPHIA, FASTER, DIVERT, and EcoTrec systems are around 0.48, 0.42, 1.93, and 1.87 s, respectively. Comparing SOPHIA and FASTER systems between each other, we can observe that the FASTER system has a slight reduction in latency. This is because the knowledge is extracted in several sub-regions, thus delivering it more rapidly to vehicles. Both SOPHIA and FASTER have an average reduction above 74% compared to the EcoTrec and DIVERT systems, as shown in the Figure 5c (bottom).

Figure 5d shows the coverage achieved as a function of vehicle insertion rate. EcoTrec has a coverage slightly larger than DIVERT because it has a lower network overhead when compared with its opponent, as depicted in Figure 5a,b. On the other hand, since FASTER and SOPHIA have lower network overloads, compared with their competitors, the knowledge extracted can reach a larger number of vehicles at all analyzed insertion rates. FASTER presents a slightly higher result compared to SOPHIA, because knowledge is extracted in several sub-regions, thus reaching coverage of 1.8% higher, see Figure 5d (bottom). There are two observations that should be considered about the development of the SOPHIA system in relation to FASTER that segments the entire scenario previously are: (i) slightly lower coverage and (ii) slightly higher latency. However, these two observations do not compromise the system’s scalability.

### 4.4. Traffic Management Assessment

This section analyzes the urban mobility management of the SOPHIA system as a function of travel time, travel time index, CO_2_ emission, and congestion time loss. The results are displayed in Figure 6. Each sub-figure is also composed of two bar charts. The top one represents the numerical value of the metric assessed and the bottom one represents the gain with respect to EcoTrec.

Figure 6a shows the result of the average travel time for all insertion rates. From the figure, it is possible to notice that the higher the vehicle insertion rate, the longer the average travel time for all solutions analyzed. This behavior is expected since, at high rates the roads become denser, leading to the occurrence of congestion. Among all solutions analyzed, EcoTrec system has the longest average travel time, around 22 min. It is known that the choice of an alternative route within it is given by the path that emits the lowest CO_2_ rate until the trip destination. Differently, the FASTER system selects a selfish route based on the probabilistic *k*-shortest path. This strategy has a gain of 6.71% agains EcoTrec. Another approach is taken by the DIVERT system, where the vehicles calculate an alternative route collaboratively. In this approach, it is possible to notice a reduction in the mean travel time around 15% and 8.3%, compared to EcoTrec and FASTER, respectively. The SOPHIA system applies collaborative routing, such as DIVERT. Even so, it overcomes DIVERT in this metric, due to the low network overhead. As mentioned before, DIVERT has a higher overhead, so many messages arrive corrupted at the recipients. Analyzing numerically, SOPHIA achieves a mean reduction of 6.46%, 14.75%, and 21.46% compared to DIVERT, FASTER, and EcoTrec, respectively, see Figure 6a (bottom).

Figure 6b indicates the level of traffic congestion as a function of vehicle insertion rate. It is observed that the results of this metric show a behavior similar to the average travel time metric (Figure 6a). This is because both metrics take into account the average travel time. As discussed earlier, the DIVERT system has a slightly higher overhead on the network compared to EcoTrec, as there are exchanges of information on alternative routes chosen by neighboring vehicles. However, this slightly higher overhead causes DIVERT to outpace its competitors (except SOPHIA) in travel time, trip time index, and two other metrics (congestion time loss and CO_2_ emission) that will be explained in more detail below.

Another important metric to be evaluated is the time lost in congestion, Figure 6c. All evaluated systems apply some vehicle rerouting mechanism after congestion detection. It is important to emphasize that systems that implement collaborative routing outperform the selfish one. This can be observed in Figure 6a,b. To demonstrate them numerically, DIVERT achieves a time reduction of approximately 7.87% and 15.14% over FASTER and EcoTrec, respectively. While SOPHIA reaches approximately 21.92% and 29.18% compared to FASTER and EcoTrec, respectively, see Figure 6c (bottom). As mentioned earlier, the SOPHIA system has a lower overhead compared to DIVERT. Therefore, this fact contributes to the information reaching the largest number of participants thus contributing to improving traffic management efficiency.

Figure 6d demonstrates the CO_2_ emission in relation to the vehicle insertion rate. As expected, EcoTrec presents the highest CO_2_ emission at all the analyzed insertion rates, since it has the highest travel time index (Figure 6b) and also the highest time lost in congestion (Figure 6c). By analyzing this metric, it is possible to observe that the most efficient systems, in the urban mobility management, present a smaller amount of CO_2_ emission. In this case, the most efficient ones are DIVERT and SOPHIA. This happens because both implement collaborative routing. Analyzing numerically, the SOPHIA, DIVERT and FASTER presented a mean reduction in CO_2_ emission, against to EcoTrec, of approximately 25.92%, 13.15%, and 5.9%, respectively, see Figure 6d (bottom).

## 5. Conclusions

There is an increasing need for efficient urban mobility management systems to improve vehicular traffic management. To meet this demand, the proposed SOPHIA system is a distributed system of urban mobility management based on a collaborative approach in vehicular social networks. The main advantage of SOPHIA is the combined use of two approaches of VSNs, such as social network concepts and analysis. A metric of social network analysis, more specifically, the egocentric betweenness metric was employed to compute the vehicle ranking. In addition, two social network concepts were employed for the collaborative route-planning, i.e., social interaction and temporal virtual community. Experimental results showed that the difficulty of selecting the most appropriate vehicle, in highly dynamic networks, can be overcome by the proposed dynamic clustering approach. The main advantage of this approach is to rely only on local knowledge of the network topology to achieve its goals. Furthermore, it was proven that SOPHIA was able to do that without jeopardizing system scalability. Another observation presented is that collaborative decision making is more efficient than selfish in alternative routes planning. In summary, the distributed solutions analyzed tend to be more scalable than the infrastructures and those that use the collaborative routing strategy are most efficient in urban mobility management.

As future work, it is intended to include the driver’s mobility patterns and user preferences in the SOPHIA system for the alternative routes planning, to personalize the alternative routes according to the driver preferences.

## Figures and Tables

**Figure 1 sensors-19-03558-f001:**
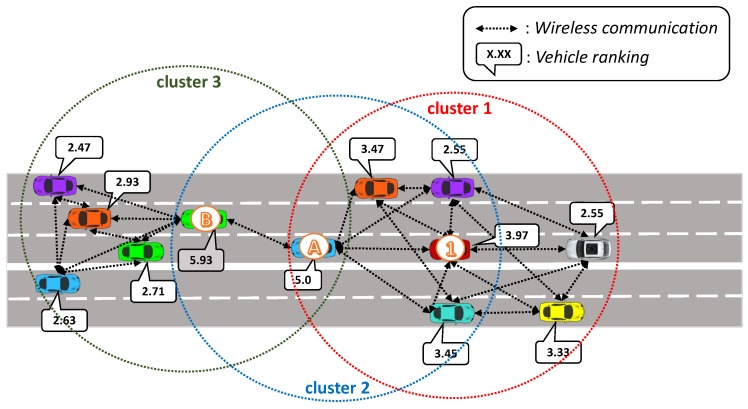
Example of clustering, the labels *A* and *B* represent the temporary CHs of groups 1 and 2.

**Figure 2 sensors-19-03558-f002:**
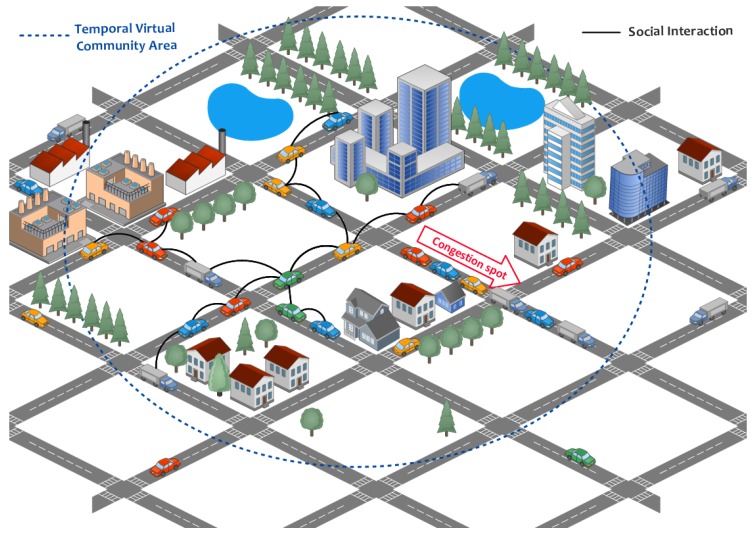
Temporal virtual community and social interactions area in VSNs.

**Figure 3 sensors-19-03558-f003:**
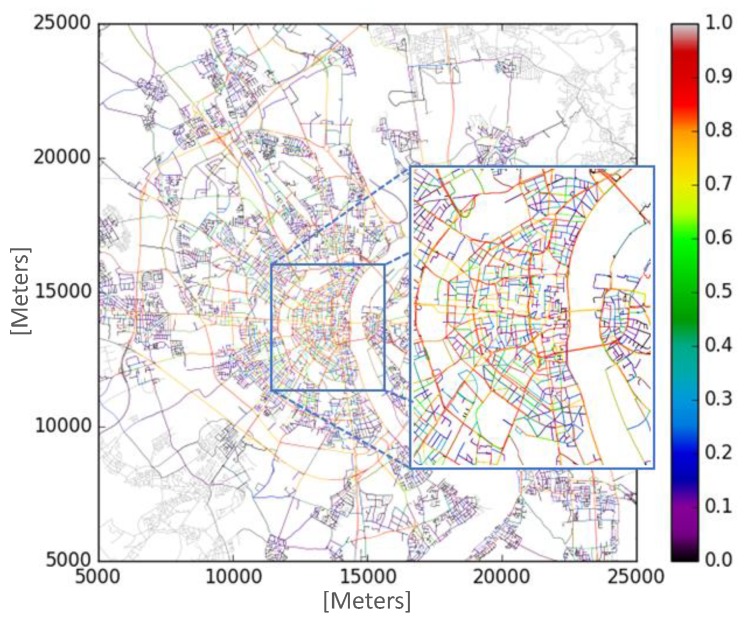
Road network of Cologne used in the simulation.

**Figure 4 sensors-19-03558-f004:**
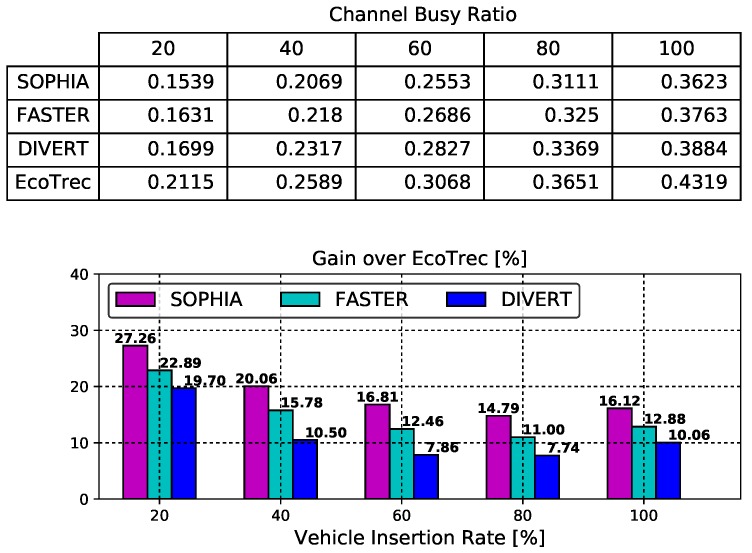
Control Channel Assessment.

**Figure 5 sensors-19-03558-f005:**
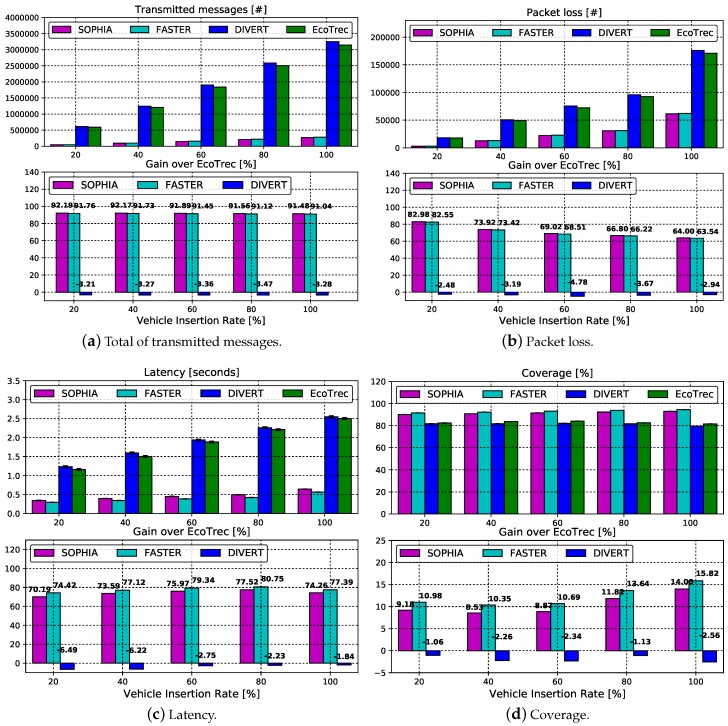
Scalability Assessment.

**Figure 6 sensors-19-03558-f006:**
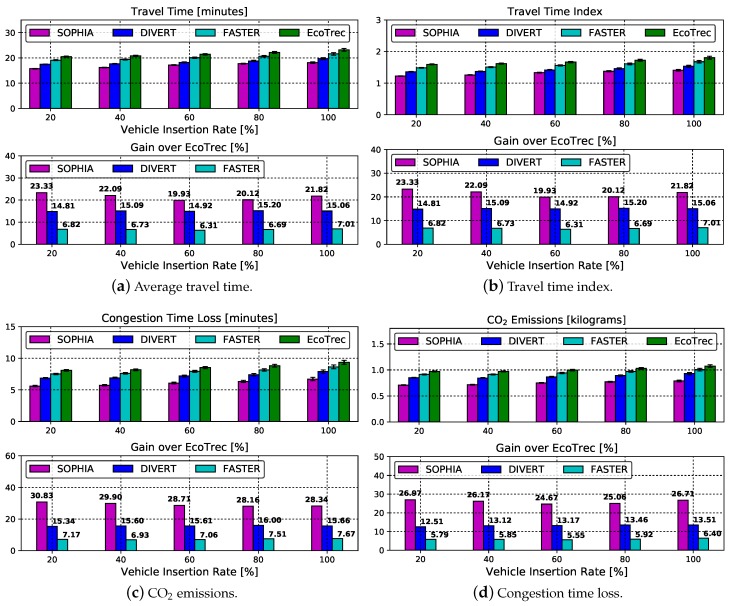
Traffic Management Assessment.

**Table 1 sensors-19-03558-t001:** Level of service and traffic classification [34].

Level of Service	Traffic Classification	$p_{i}$
A	Free flow	(0.0∼0.33]
B	Reasonably free flow	(0.33∼0.4]
C	Stable flow	(0.4∼0.5]
D	Approaching unstable flow	(0.5∼0.7]
E	Unstable flow	(0.7∼0.9]
F	Forced or breakdown flow	(0.9∼1.0]

**Table 2 sensors-19-03558-t002:** Simulation parameters settings.

Parameter	Value
Vehicle Insertion Rate	20% to 100%
MAC layer	IEEE 802.11p PHY
Bandwidth	10 MHz
NIC Bitrate	6 Mbps
NIC TX power	20 mW
NIC Sensitivity	-82 dBm
Transmission range	287 m
Beacon transmission rate	1 Hz
Confidence interval	95%

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
