# Peer review of "Exploiting Vehicular Social Networks and Dynamic Clustering to Enhance Urban Mobility Management"

_sensors, 2019, doi:10.3390/s19163558_

Round 1

Reviewer 1 Report

A distributed approach to provide an efficient mobility management system for urban mobility is proposed using social network concepts. The paper is well written and presents detailed information. A minor comment is as below,

The dynamic clustering approach uses EBM metric (authors previous work), however, how does the dynamic clustering care take the vehicles present in multiple clusters?

Author Response

The authors would like to thank the reviewer for spending his/her time assessing the manuscript and for the constructive and detailed comments provided.

Please refer the attachment.

Reviewer 2 Report

Exploiting Vehicular Social Networks and Dynamic Clustering to Enhance Urban Mobility Management

The text shows good reading flow. However, it contains some little English grammar issues that need careful attention. Please refer to the list below as is contains some examples of the issues found in text during the reading:

1. Verbs
1.1 "Thus, increasing network scalability and decreasing communication overhead." -> missing verb

2. Commas
2.1 "ones such as vehicular" -> "ones, such as vehicular"
2.2 "vehicular environment such as"
2.3 "congested areas, but rather only" -> "congested areas but rather only"
2.4 "thus, eliminating the difficult task" -> "thus eliminating the difficult task"
2.5 "channel busy ratio, while the bar chart"
2.6 "thereby, raising the competition"

3. Confusing:
3.1 "Beyond that, two communication types most commonly applied in VSNs are vehicle-to-vehicle (V2V) and vehicle-to-infrastructure (V2I) [9]." -> disconnected

4. Pronouns:
4.1 "This is performed together" -> To what is "This" referring?
4.2 "This will include the aggregate information received from"

5. Articles:
5.1 "the vehicle B" -> "vehicle B"
5.2 "the Algorithm 2 is"

6. "and so on." -> vague

7. Use, sometimes, a shorter and well-accepted construct for "In order to": "To" -> For instance, you change "In order to overcome this challenge" to just "To overcome this challenge"

8. It needs a more objective text style. The text gives details incrementally; it becomes quite repetitive and hampers the reading.

9. Try avoiding the use of "that is" and "in other words". Avoid additional explanations for the same term - describe/explain it just once.

In what regards the content of the paper, there are some comments that authors must really consider for correcting their manuscript (to be suitable for publication):

I suggest the authors to be more objective when introducing their proposal in the Introduction -> "Based on the gaps found, (...)" referring to the enlisted issues is quite generic.

Since the proposed approach is directly targeting the scope of Subsection 2.3, it would be interesting to have an additional paragraph that tells/summarizes the issues (gaps) that the previous works undergo and SOPHIA is fixing.

In Equation 1, why is only location pair up with time stamp instead with all other parameters in the tuple?

It would be interesting to add an in-place explanation of what an "infrastructural approach" is for the statement "A straight solution for this problem is to employ an infrastructural approach [10–12], (...)"

The authors state that "For each change in the local topology, the vehicle’s score should be updated." What do they mean by CHANGE in the local topology?

What would it be the cost of Cluster Management for supporting SOPHIA?

In Subsection 3.3, vi is a vertex/intersection and vmi is the average speed of an edge/segment. I would recommend changing the name of variables better clarity.

Table 1 refers to wi as the weight while the text refers to the weight as (p) (Equation 5).

Is Table 1 a fuzzification (discrete conversion) of traffic conditions?

The work needs further explanation and contextualization about social aspects, such as "road popularity".
Is Equation 6 just considering the neighbouring vehicles' road segment choices for a given instant?
What do the authors mean by "road"? Is it just "road segment" or a set of straight "road segments"?
Why is the length of a road directly proportional to its popularity?

It is nice to have an extensive performance analysis with 9 metrics? However, would not only type 3 metrics important for the performance of SOPHIA (the only ones that really matter)?
Maybe using a metric derived from popularity (Equation 6) would also give an interesting perspective of performance where the approach's performance relates to the overall average road popularity on the map (or the least variance of the popularities).

Is not the lowest Channel Busy Ratio by SOPHIA brought by the individualism of the protocol?

In Figure 6, the captions of the graphs are misplaced.

In what regards the traffic management analysis, are not most of (or all) the metrics tightly correlated? Improving travel time will show better Travel time index and CO2 Emissions?

Again, showing the overall traffic load distribution (popularity) would actually show how well this fully distributed approach is performing in terms of balancing traffic load over the urban area.

I would strongly recommend the authors to explore a saturation (heavy traffic or high density) scenario.

Author Response

(The authors gave the same response as above.)

Reviewer 3 Report

 In this paper, authors propose a collaborative approach in vehicular social networks (VSNs), named SOPHIA. The VSN paradigm emerged from the integration of mobile communication devices and their social relationships in the vehicular environment. Thereby, social network analysis (SNA) and social network concepts (SNC) are two approaches that can be explored in VSNs. Their proposed solution adopts both SNA and SNC approaches for alternative route planning in a collaborative way. Additionally, they used dynamic clustering in order to select the most appropriate vehicles in a distributed manner. Simulation results confirmed that the combined use of SNA, SNC, and dynamic clustering, in the vehicular environment, have great potential in increasing system scalability as well as improving urban mobility management efficiency.

After reading the paper, this reviewer considers that the paper covers an interesting issue and it is well motivated, so I recommend accepting this work. 

Author Response

(The authors gave the same response as above.)
